# Ketone α-alkylation at the more-hindered site

Ming-Ming Li[1], Tianze Zhang[1], Lei Cheng[1], Wei-Guo Xiao[1], Jin-Tao Ma[1], Li-Jun Xiao [1]✉ & Qi-Lin Zhou [1]✉

Control of the regioselectivity of α-alkylation of carbonyl compounds is a longstanding topic of research in organic chemistry. By using stoichiometric bulky strong bases and carefully adjusting the reaction conditions, selective alkylation of unsymmetrical ketones at less-hindered α-sites has been achieved. In contrast, selective alkylation of such ketones at more-hindered α-sites remains a persistent challenge. Here we report a nickel-catalysed alkylation of unsymmetrical ketones at the more-hindered α-sites with allylic alcohols. Our results indicate that the space-constrained nickel catalyst bearing a bulky biphenyl diphosphine ligand enables the preferential alkylation of the more-substituted enolate over the less-substituted enolate and reverses the conventional regioselectivity of ketone α-alkylation. The reactions proceed under neutral conditions in the absence of additives, and water is the only byproduct. The method has a broad substrate scope and permits late-stage modification of ketone-containing natural products and bioactive compounds.

α-Alkylation of carbonyl compounds is one of the most commonly used carbon–carbon bond forming reactions in organic synthesis. In retrosynthetic analyses, α-alkylation of a carbonyl compound is the connection between an enolate of the carbonyl compound and an alkylating agent[1]. Because most ketones are unsymmetrical (i.e., they have two different alkyl groups), two regioisomeric enolates can be formed upon deprotonation (Fig. 1a), which can lead to the mixture of structurally isomeric alkylated products, and thus controlling the regioselectivity of α-alkylation of unsymmetrical ketones is a longstanding topic of research in complex molecule syntheses[1–4], as documented in almost all organic chemistry textbooks[1]. In the past half a century, remarkable progress has been achieved in the selective alkylation of unsymmetrical ketones at their less-hindered α-sites. One conventional strategy is to use a stoichiometric amount of a bulky strong base (e.g., lithium diisopropylamide or potassium bis(trimethylsilyl)amide) in an aprotic solvent under cryogenic conditions to generate the kinetically favored less-substituted enolates[1–6] (Fig. 1a). Another strategy involves the Stork enamine reaction, which yields products of alkylation at the less-hindered α-sites via preformation of

less-substituted enamines[7,8]. In addition, several elegant examples of metal–amine cooperative catalysis have been reported for the alkylation of carbonyl compounds by means of the indirect enamine strategy[9–15].

However, methods for selective alkylation of unsymmetrical ketones at their more-hindered α-sites remain a challenge. This difficulty escalates when less-substituted enolate is both kinetically and thermodynamically favored over more-substituted enolate (Fig. 1a). To date, many efficient strategies devised to overcome this problem including ketone pre-activation with stoichiometric aluminum tris(2,6-diphenylphenoxide)[16] and preformations of silyl enol ethers[16,17], enol carbonates[18], and β-ketoesters[19]. However, despite the significant progress for selective alkylation at more-hindered α-sites of unsymmetrical ketones, these established methods still suffer from the tedious prefunctionalization steps, cryogenic manipulations, stoichiometric use of metallic bases, and the unwanted formation of stoichiometric salt wastes. There are only rare examples of direct alkylation of unsymmetrical cycloketones at their more-hindered α-site, including the allylation of 2-methylcyclohexanone by using

[1]State Key Laboratory and Institute of Elemento-Organic Chemistry, College of Chemistry, Frontiers Science Center for New Organic Matter, Nankai University, Tianjin 300071, China. ✉e-mail: ljxiao@nankai.edu.cn; qlzhou@nankai.edu.cn

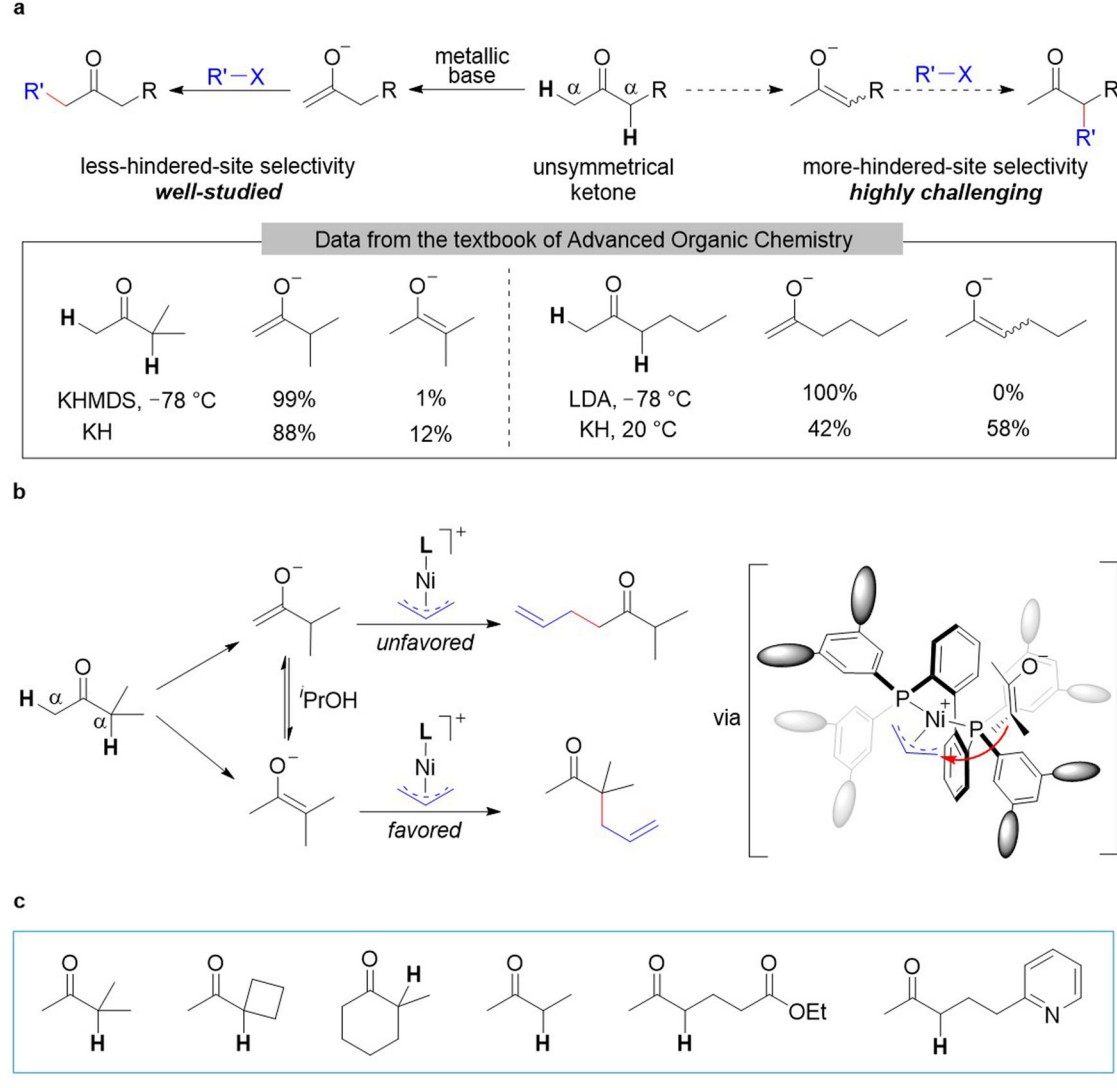

**Fig. 1 | Regioselective alkylation of unsymmetrical ketones. a** Conventional alkylation of ketones and composition of enolate mixtures generated by various bases. LDA lithium diisopropylamide, KHMDS potassium bis(trimethylsilyl)amide. **b** Proposed strategy for selective alkylation reaction at the more-hindered α-site. L ligand. **c** Scope of this allylic alkylation reaction.

phosphoric acid catalysts[20–22]. Therefore, it is highly desirable to develop straightforward and general protocols for regioselective alkylation at the more-hinder α-sites of unsymmetrical ketones.

Transition-metal-catalysed allylic alkylation of ketones has been well-studied[23–28]. However, research in this area has focused mainly on ketones that form stabilized enolates[23–30], preformed silyl enol ethers[31,32], and the alkyl silyl ketones[33]. Recently, seminal work by the groups of Mashima[34,35], Sauthier[36], Stoltz[37], and Zhang[35] has shown that nickel-π-allyl intermediates can undergo coupling reactions with stabilized carbon nucleophiles. We wondered whether we could, by designing a suitable ligand, control the regioselectivity of coupling reactions between a nickel-π-allyl intermediate and one of the two enolates of unsymmetrical ketones, which are in equilibrium in protonic solvents. This well-designed catalyst would selectively recognize the more-substituted enolate and react with it preferentially (Fig. 1b). Here, we report that a bulky bis(triarylphosphine) ligand (DTBM-BP) enabled us to carry out nickel-catalyzed allylic alkylation of unsymmetrical dialkyl ketones and allylic alcohols selectively at the more-hindered α-sites of the ketones, without the need for a base or any additives and with water as the only byproduct. This catalytic method was applicable to various unsymmetrical ketones (Fig. 1c).

## Results and discussion

### Optimization of reaction conditions

We began by investigating allylic alkylation reactions of the dialkyl ketone 3-methylbutan-2-one (**1a**) and (*E*)−3-(4-methoxyphenyl) prop-2-en-1-ol (**2a**) as model substrates. Various diphosphine ligands were systematically screened in EtOH at 80 °C (Fig. 2). Commonly used diphosphine ligands such as DPPB, DPPMB, BINAP, DPPF, and Xantphos either failed to afford desired allylic alkylation product **3a** or gave low yields (<35%). However, when the biphenyl diphosphine ligand Ph-BP (**L1**) was used, the yield of **3a** slightly increased (to 40%), and the regioisomeric ratio (r.r., **3a/4a**) was high (95:5). We were delighted to find that the yield and regioselectivity were significantly improved by fine-tuning the aryl substituents on the ligand. Increased yields and higher regioselectivities were achieved with 3,5-dimethyl-substituted (Xyl-BP, **L4**, 62%, r.r. = 98:2), 3,5-di(*tert*-butyl)-substituted (DTB-BP, **L5**, 71%, r.r. > 99:1), and 3,5-di(*tert*-butyl)−4-methoxy-substituted (DTBM-BP, **L6**, 81%, r.r. > 99:1) ligands, indicating that the bulky biphenyl diphosphine ligands are favorable for the reaction. In addition, experiments with ligands having different electric properties showed lower yields for ligands with electron-deficient substituents (4-fluorophenyl, **L2**) and higher yields for ligands with electron-donating substituents

**Fig. 2 | Optimization of ligand for nickel-catalysed regioselective allylic alkylation of unsymmetrical ketones.** Reaction conditions: Ni(COD)$_2$ (10 mol%), ligand (11 mol%), EtOH (0.5 mL), ketone **1a** (0.1 mmol), allylic alcohol **2a** (0.1 mmol), 80 °C, 12 h. Yields were determined by $^1$H NMR spectroscopy. Regioisomeric ratios (r.r., **3a/4a**) were determined by gas chromatography. N.D. not detected, PMB 4-methoxybenzyl, COD 1,5-cyclooctadiene.

(4-methoxyphenyl, **L3**) compared to ligand **L1** (phenyl). However, the electric property of the ligand has no effect on the regioselectivity of the reaction.

## Substrate scope

Using **L6** as the ligand, we explored the scope of the reaction with respect to the unsymmetrical dialkyl ketone substrate (Fig. 3). First, we tested ketones with primary and tertiary α-carbons (Fig. 3a). Because of the alkyl/alkyl steric repulsion on the same side of the more-substituted enolates of such ketones, less-substituted enolate is both kinetically and thermodynamically favored over more-substituted enolate. Nevertheless, reactions of ketones **1a–1g** with **2a** catalysed by our nickel catalysts smoothly produced **3a–3g**, the products of monoallylation at the tertiary α-carbon, in moderate to high yields (51–81%) with excellent regioselectivities (r.r. > 99:1). Notably, the method was feasible for methyl cycloalkyl ketones bearing a 4-, 5-, 6-, or 7-membered ring, affording products **3c–3g**.

Next, we studied the ketones with secondary and tertiary α-carbons (Fig. 3b). 2-Alkyl-substituted cyclic ketones 2-methylcyclopentanone, 2-methylcyclohexanone, and 2-propylcyclohexanone underwent allylic alkylation at their tertiary α-carbons to afford allylation products **3h–3k**, respectively, in high yields (68–80%) with excellent regioselectivities (r.r. > 99:1). Notably, a nitrogen atom in the ring of a cyclic ketone had no effect on the reaction outcome, affording **3k** in 74% yield. However, the reaction of **1l**, an open-chain ketone with secondary and tertiary α-carbons, showed low regioselectivity (r.r. = 2.3:1).

We then tested a series of ketones **1m–1z** with primary and secondary α-carbons (Fig. 3c). However, under the same conditions as the ketones with secondary and tertiary α-carbons, the allylation reaction of ketone **1m** gave the desired product **3m** in only 33% yield. When the amount of ketone substrate was increased to three equivalents and the solvent was changed to $^i$PrOH, the yield of **3m** increased to 72% (see Supplementary Fig. 1 in Supplementary Information). Under the optimal condition (Supplementary Fig. 1, entry 5 in Supplementary Information), the monoallylation products at the secondary α-carbon were obtained in moderate to high yields (52–83%). And the regioselectivity was >99:1 for all tested ketones containing primary and secondary α-carbons. Even the simplest dialkyl ketone, 2-butanone, was exclusively monoallylated at the more-substituted α-site to afford **3n** in good yield with excellent regioselectivity. Amine (**3s**), halogen (**3t**), ether (**3u**), olefin (**3v**), and amide (**3w**) substituents were tolerated in the reaction. Notably, the presence of heterocyclic rings in the ketone substrates has no effect on the reaction: desired monoallylation products **3x–3z** were obtained in 54–68% yields.

We next studied the scope of the reaction with respect to the allyl alcohol and were pleased to find that various alcohols were suitable (Fig. 4). Regardless of whether linear or branched alcohol was used, the allylation reaction occurred at the terminal carbon of the allyl group. For example, the same product was obtained from the reactions of allylic alcohols **5a** and **5g** with 2-methylcyclopentanone (**1h**), indicating that both reactions proceeded via the same nickel-π-allyl intermediate[37]. Both aromatic and aliphatic allylic alcohols reacted

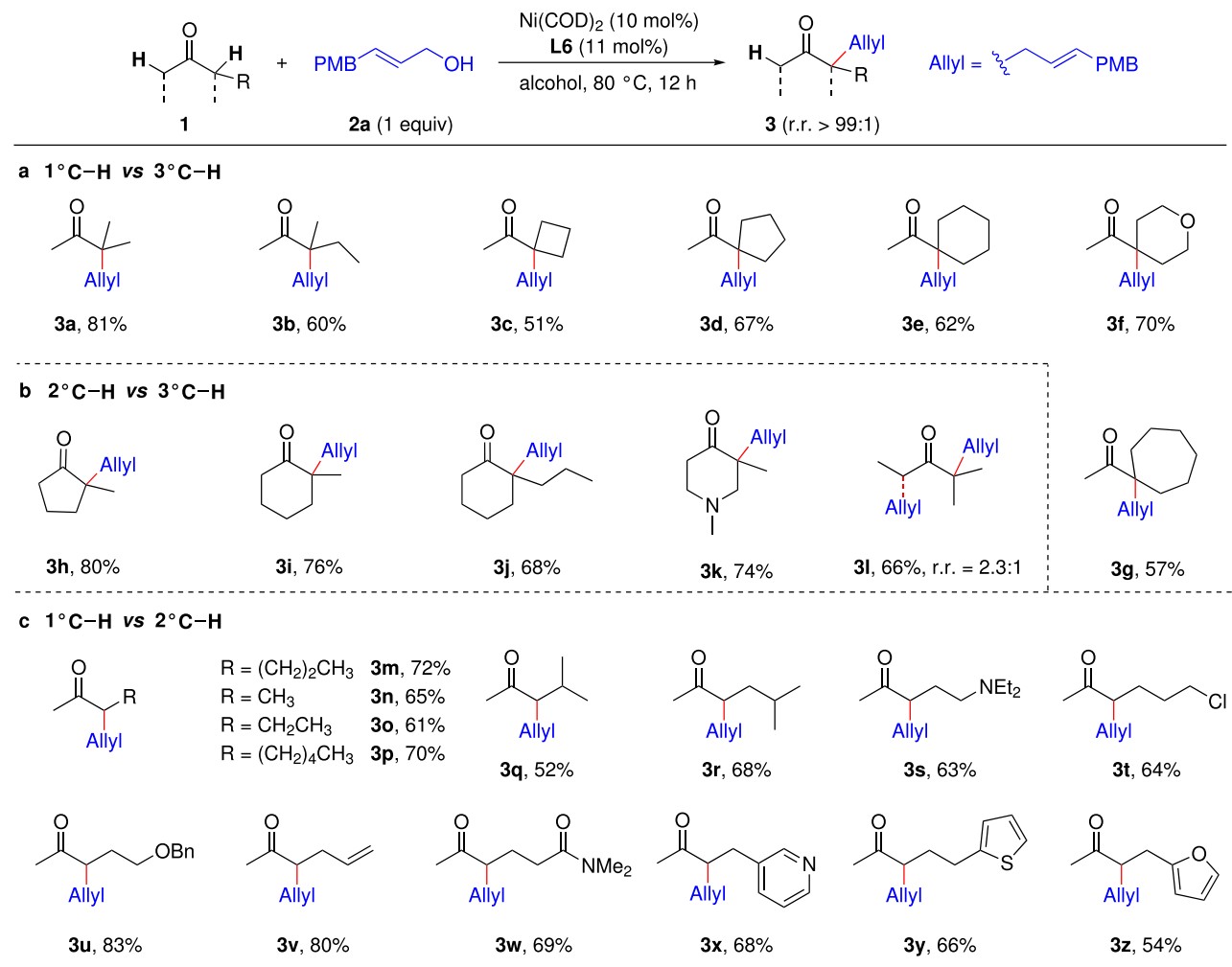

**Fig. 3 | The scope of unsymmetrical ketones.** Reaction conditions: Ni(COD)₂ (10 mol%), **L6** (11 mol%), EtOH (0.5 mL), ketone **1** (0.1 mmol), allylic alcohol **2a** (0.1 mmol), 80 °C, 12 h. Isolated yields are given. r.r. > 99:1 unless noted. **a** Ketones with primary and tertiary α-carbons. **b** Ketones with secondary and tertiary α-carbons. **c** Ketones with primary and secondary α-carbons. Ketone **1** (0.3 mmol), allylic alcohol **2a** (0.1 mmol). Using ʲPrOH as solvent.

with **1h** to give allylation products **6a**–**6j** in 51–80% yields with excellent regioselectivities (r.r. > 99:1). A good result was obtained even with the simplest allylic alcohol (**5k**). However, prenol and geraniol are inert in this reaction. It is noteworthy that the use of a 1:1 mixture of *E*- and *Z*-allylic alcohols yielded only a product with an (*E*)-olefin (**6b**). In addition to allylic alcohols, an allylic ether (**5l**), amine (**5 m**), or ester (**5n**) could be used as an allylation reagent[38].

## Applications

Because cyclic ketones bearing a quaternary α-carbon stereocenter are found in many naturally occurring compounds and pharmaceuticals and are also versatile building blocks in organic synthesis[39], we performed a gram-scale preparation of cyclic ketone **6a**, which has been widely used in the synthesis of carbocycles[40]. Notably, in this gram-scale reaction, the catalyst loading could be reduced to 7 mol% with no decrease in the yield or regioselectivity (Fig. 5a), thus demonstrating the synthetic utility of the method.

When a ketone has not only its two α-sites but also another active site that can be alkylated, selective monoalkylation becomes even more challenging. To assess both the regioselectivity and the chemoselectivity of our method, we carried out allylic alkylations of ketones **1aa** and **1ab**, which have three possible reaction sites. In these experiments, allylation occurred only at the secondary α-carbons of the ketones (Fig. 5b). The chemoselectivities of these reactions may be attributed to the pKₐ value of the secondary C–H bonds: C–H bonds

with lower pKₐ values are preferentially alkylated (see Supplementary Fig. 5 in Supplementary Information).

Because unsymmetrical ketones are abundant in natural products and bioactive molecules, methods for direct regioselective alkylation reactions are of great significance for expanding the diversity of bioactive compounds and discovering new drugs. Our allylic alkylation reactions of ketones are carried out under neutral conditions in alcohol without the use of bases or additives, and therefore we expected the method to be useful for the late-stage modification of ketone-containing natural products and bioactive molecules. With that in mind, we carried out allylic alkylations of a variety of molecules, including dihydrocarvone, pentoxifylline, ursodeoxycholic acid derivative, and abiraterone derivative (Fig. 5c). All the studied molecules underwent allylation selectively at the more-substituted α-sites of the ketones to afford desired products **3ac**–**3af** in moderate to good yields with excellent regioselectivities (r.r. > 99:1).

## Mechanism studies

To gain insight into the mechanism of the reaction, we performed deuterium-labeling experiments. First, the reaction of **1ag** was carried out in isopropanol-*d₈* to probe the keto–enol tautomerization of the substrate (Fig. 6a). In the absence of allylic alcohol, no H/D exchange at the α-sites of the ketone was observed. However, in the presence of allylic alcohol, both substrate **1ag** and product **3ag** were partially deuterated at their α-sites. The rates of deuteration on the two α-sites

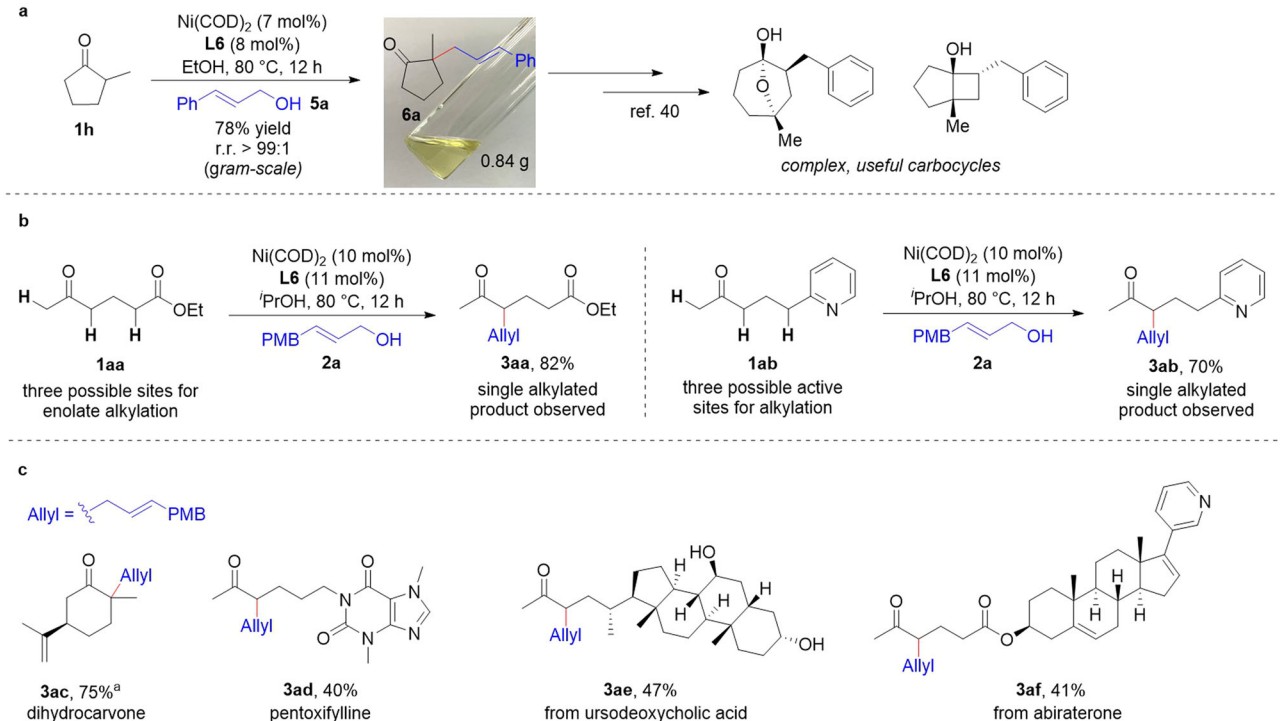

**Fig. 4 | The scope of allylation reagents.** Reaction conditions: Ni(COD)₂ (10 mol%), **L6** (11 mol%), EtOH (0.5 mL), ketone **1h** (0.1 mmol), allylic alcohol **5** (0.1 mmol), 80 °C, 12 h. Isolated yields are given. r.r. > 99:1 unless noted. ᵃUsing a 1:1 mixture of *Z*- and *E*-3-phenylprop-2-en-1-ol. ᵇThe yield was measured by GC analysis; r.r. = 97:3.

**Fig. 5 | Applications of nickel-catalyzed allylic alkylation of unsymmetrical ketones. a** Regioselective allylic alkylation on a gram scale. Ketone **1h** (5.0 mmol), allylic alcohol **2a** (5.0 mmol). **b** Regioselective and chemoselective allylic alkylation of ketones. Ketone (0.3 mmol), allylic alcohol **2a** (0.1 mmol). **c** Late-stage modification of natural and bioactive ketones. Reaction conditions: Ni(COD)₂ (10 mol%), **L6** (11 mol%), ketone (0.1 mmol), allylic alcohol (0.1 mmol), ᶦPrOH (0.5 mL). ᵃUsing EtOH as solvent.

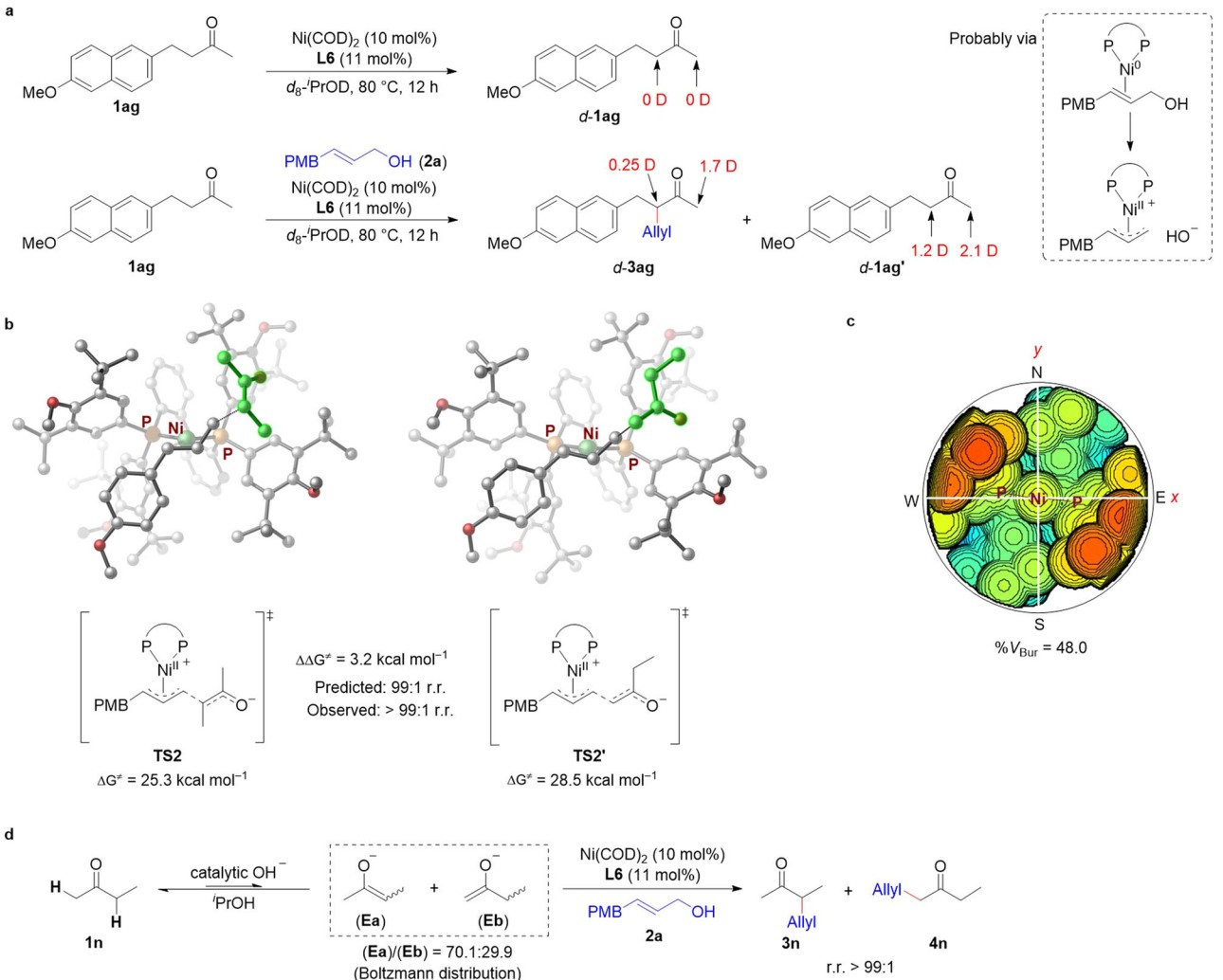

**Fig. 6 | Mechanism Study of the Allylic Alkylation Reaction of Ketones.**
**a** Deuterium-labeling experiments. **b** Transition-state structures of **TS2** and **TS2'** by DFT. Single-point energies were calculated at the PBE0-D3(BJ)/def2-TZVPP/SMD-($^i$PrOH) level of theory with structures optimized at the B3LYP-D3(BJ)/SDD(Ni) +6−31 G(d)/SMD-($^i$PrOH) level. The ligand used for the calculations is **L6**. **c** Steric map of the catalyst Ni(II)/**L6** on the basis of the DFT-optimized structure of **TS2**. The steric map is viewed down the *z*-axis; the orientation of the catalyst Ni(II)/**L6** is the same as **TS2**. The red and blue zones indicate the more-hindered and less-hindered zones in the catalytic pocket, respectively. $\%V_{Bur}$, percentage of buried volume. **d** The ratio of two kinds of enolates (**Ea**) and (**Eb**) of 2-butanone was calculated under the Boltzmann distribution assumption.

of recovered *d*-**1ag'** were 60% (1.2 D) and 70% (2.1 D), respectively. These results indicate that the allylic alcohol reacted with the zero-valent nickel catalyst to generate a catalytic amount of OH⁻, which promoted keto−enol tautomerization.

We studied the origin of the regioselectivity by means of density functional theory (DFT) calculations using the reaction of butan-2-one **1n** with allylic alcohol **2a** as a model (Supplementary Fig. 8 in Supplementary Information). We calculated two possible pathways, one leading to allylation at the less-hindered α-site and one to allylation at the more-substituted α-site. The calculations showed that C−C bond formation has the highest energy barrier and is thus the rate-determining step. The energy of the transition state for allylation at the more-substituted α-site (**TS2**) is 3.2 kcal/mol lower than that of the transition state for allylation at the less-substituted α-site (**TS2'**) (Fig. 6b). This result indicates that the more-substituted enolate preferentially attacks the cationic nickel-π-allyl intermediate. As depicted in Fig. 6c, the 3D structure and steric map of the catalyst Ni/**L6** shows that the sterically demanding bulky aryl substituents on the phosphine atoms constitute a space-constrained and deep pocket, which can differentiate the two isomeric enolates (Supplementary Fig. 9,

Supplementary Fig. 10 in Supplementary Information). In addition, using the Boltzmann distribution assumption, we calculated the ratio of the two enolates of 2-butanone (**1n**) under the reaction conditions and found that the ratio of the more-substituted enolate (**Ea**) to the less-substituted enolate (**Eb**) is only 70.1:29.9 (Fig. 6d and Supplementary Table 4 in Supplementary Information). Therefore, the nickel catalyst that recognizes the more-substituted enolate and stabilizes the transition state for C−C bond formation at the more-hindered α-site is responsible for the high regioselectivity of the reaction. In addition, we also calculated the transition barriers by taking chain ketone **1l** as a model. The DFT calculation results indicated that the transition barrier difference between the alkylations at more-substituted α-site and less-substituted α-site is only 1.1 kcal/mol, which is consistent with the observed low regioselectivity (Supplementary Table 5 in Supplementary Information).

In summary, we have developed a method for highly regioselective nickel-catalysed allylic alkylation at the more-substituted α-sites of unsymmetrical dialkyl ketones. The use of a bulky biphenyl diphosphine ligand is the key to controlling regioselectivity. The reactions take place under neutral conditions without any additives, and water is

the sole byproduct. The method has a broad substrate scope and is suitable for late-stage modification of ketone-containing natural products and bioactive molecules. Mechanistic studies showed that the space-constrained nickel catalyst reacts with the allylic alcohol to generate a nickel-π-allyl intermediate and OH⁻, which promotes enolate formation from the ketone; and the more-substituted enolate preferentially attacks the nickel-π-allyl intermediate to form the desired allylation product.

## Methods

### General procedure for catalytic regioselective allylic alkylation of ketone 1a with allylic alcohol 2a

Ni(COD)$_2$ (2.8 mg, 10 mol%), **L6** ligand (12 mg, 11 mol%), ketones **1a** (9 mg, 0.1 mmol), and allyl alcohols **2a** (16 mg, 0.1 mmol), in that order, were placed in a reaction tube with a magnetic stir bar under argon. After EtOH (0.5 mL) was injected into the tube, the reaction mixture was stirred at 80 °C for 12 h (600 rpm). On completion, the reaction mixture was cooled to room temperature. Organic volatiles was evaporated in vacuo, and the residue was purified by preparative thin-layer chromatography to afford the desired allylated ketone **3a** (19 mg, 81% yield, >99:1 r.r.).

## Data availability

All data regarding materials and methods, optimization studies, experimental procedures, DFT calculations, and NMR spectra can be found in the Supplementary Information. The Cartesian coordinates of the optimized structures are included in the Supplementary Data. All other data are available from the corresponding authors upon request.

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

## Acknowledgements

We thank the National Key R&D Program of China (2022YFA1504302), the National Natural Science Foundation of China (Nos. 91956000 and 22188101), the Fundamental Research Funds for the Central Universities, and the Haihe Laboratory of Sustainable Chemical Transformations for financial support.

## Author contributions

Q.L.Z. conceived the study; L.J.X. and Q.L.Z. guided the study; M.M.L., L.J.X., and Q.L.Z. designed the experiments and analyzed the data; M.M.L. performed the reactions and mechanistic studies; T.Z. performed the DFT calculations; L.C., W.G.X., and J.T.M. made some of the ketone substrates; L.J.X. and Q.L.Z. wrote the paper.

## Competing interests

The authors declare no competing interests.
