## [Peer Review File · Nature Communications]

REVIEWER COMMENTS

Reviewer #1 (Remarks to the Author):

In this manuscript, Zhou, Xiao and co-workers reported a novel nickel-catalysed alkylation of unsymmetrical ketones with allylic alcohols. With the assistance of bulky phosphine ligand, an excellent regioselectivity has been achieved at the more-substituted α -sites of unsymmetrical ketones. The reaction proceeded smoothly under simple conditions, showing good substrate tolerance. In addition, deuterium-labeling experiments and density functional theory (DFT) calculations gave a reasonable explanation for the regioselectivity. Thus, this reviewer recommends it to be published after minor revision.

1) Please check the sentence "Here we report that a bulky bis(triarylphosphine) ligand (DTBM-BP) enabled us to carry out nickel-catalyzed allylic alkylation of unsymmetrical dialkyl ketones and allylic alcohols selectively at the more-hindered α -sites of the ketones, without the need for a base or any additives and with water as the only byproduct. This catalytic method was applicable to various unsymmetrical ketones (Fig. 1c)." If this sentence is misplaced?

2) During the past years, Palladium has been proved as powerful catalyst for allylic alkylation, so how about the results when palladium was used in this reaction?

3) In generally, most of the products were obtained in 50-70% yields, if any other by-products are detected or the starting materials are recovered?

Reviewer #3 (Remarks to the Author):

Zhou and colleagues reported an intriguing reactivity of the Ni/P-P catalytic system for α -allylation of ketone at the sterically hindered position over the less sterically hindered favourable site. Because the reaction does not require any base, the only byproduct is water. A good scope of allylic alcohols and excellent regioselectivity may lead to this method being useful to a broader community. The linear selectivity obtained from branched alcohol is intriguing, indicating that the reaction proceeds via Ni- α -allyl species. The manuscript can be considered for publication; however, before acceptance, the following questions must be addressed.

1) Do other types of alcohols (not just allylic) react in the same way? Consider 1-octanol as an example. In doing so, we will rule out the alternative allylation pathway involving hydrogen borrowing chemistry.

2) It would be fascinating if the author included terpenols (like geraniol, nerol, geranylgeraniol, etc.) in the substrate scope. This will be the first step in applying this method in the targeted synthesis.

3. How is stereoselectivity transformation accomplished? The question is why it is E-configuration rather than a combination of E and Z. If a mixture of E and Z allylic alcohols (1:1) is subjected to the reaction conditions, what would happen?

This article by Zhou and co-workers demonstrates regioselective control towards α -alkylation of ketones at more hindered site. They have screened several diphosphine chelating ligand with Ni(COD)₂ to get the best protocol for regioselective α -alkylation at the hindered site. Overall, this approach makes an important improvement on the aspect of alkylation of a ketone at its hindered site. This reviewer supports publication in Nature Comm if the queries are addressed.

Comments :

- (1) There is a very clear trend that electron-donating substituents are performing better in terms of overall yield, although the regioselectivity does not change much. Is it possible to comment on the trend of the ligand substitutions? Authors should perform some kinetic experiments to see whether these substituents influence the oxidative addition step of the allylic alcohol.
- (2) Line 98: The authors reported low regioselectivity for an open chain ketone **11** with secondary and tertiary α -carbons. Can the authors compute the transition barriers taking **11** as the model to compare the transition states and explain the low regioselectivity accordingly.
- (3) Figure 6a: Can the authors explain the results of deuterium labelling experiments? Why percentage of deuterium incorporation is higher at the less-substituted end as compared to the more-substituted end, although the ratio of the more-substituted enolate to the less-substituted enolate is 70.1:29.9 (line 178).
- (4) Figure 5b: Authors mentioned regioselective and chemoselective allylic alkylation of ketones. Is that due to higher pka of those C-H's, since keto-enol tautomerism might not be possible then with the catalytic OH⁻. Some substrate having C-H with lower pka should be tried to check whether this chemoselectivity is a pka dependent phenomenon or not.
- (5) Line 167: The figure references should be placed where it has been discussed.
- (6) The authors should explain the significance of the data shown in table S2.
- (7) Table S3: Why does TS1'' poses that high energy as compared to TS1? Why does the involvement of two molecules of isopropanol increase the transition barrier? Is it due to entropy-factor or space-constraint of the catalytic pocket?
- (8) Table S6: What is the difference between the optimized geometries E_b and E_b'? Is that a local minimum or that has been done through constraint optimization? That should be checked by frequency calculations and mentioned clearly.

REVIEWER COMMENTS

Reviewer #1 (Remarks to the Author):

In this manuscript, Zhou, Xiao and co-workers reported a novel nickel-catalysed alkylation of unsymmetrical ketones with allylic alcohols. With the assistance of bulky phosphine ligand, an excellent regioselectivity has been achieved at the more-substituted α -sites of unsymmetrical ketones. The reaction proceeded smoothly under simple conditions, showing good substrate tolerance. In addition, deuterium-labeling experiments and density functional theory (DFT) calculations gave a reasonable explanation for the regioselectivity. Thus, this reviewer recommends it to be published after minor revision.

1) Please check the sentence “Here we report that a bulky bis(triarylphosphine) ligand (DTBM-BP) enabled us to carry out nickel-catalyzed allylic alkylation of unsymmetrical dialkyl ketones and allylic alcohols selectively at the more-hindered α -sites of the ketones, without the need for a base or any additives and with water as the only byproduct. This catalytic method was applicable to various unsymmetrical ketones (Fig. 1c).” If this sentence is misplaced?

Response: Thanks for pointing out the mistake. The sentence has been moved to the end of paragraph 3 of introduction in the revised manuscript.

2) During the past years, Palladium has been proved as powerful catalyst for allylic alkylation, so how about the results when palladium was used in this reaction?

Response: When the palladium catalysts, such as Pd(OAc)₂ and Pd₂(dba)₃, were used, no desired product was obtained. This information has been added in the revised SI.

3) In generally, most of the products were obtained in 50-70% yields, if any other by-products are detected or the starting materials are recovered?

Response: In some cases, the starting materials are recovered, and in most cases, allyl etherification by-products are detected. The conversion of substrate and by-product yield of the reaction of **1m** are added in the revised SI (Table S1).

Reviewer #2 (Remarks to the Author):

This article by Zhou and co-workers demonstrates regioselective control towards α -alkylation of ketones at more hindered site. They have screened several diphosphine chelating ligand with Ni(COD)₂ to get the best protocol for regioselective α -alkylation at the hindered site. Overall, this approach makes an important improvement on the

aspect of alkylation of a ketone at its hindered site. This reviewer supports publication in Nature Comm if the queries are addressed.

1) There is a very clear trend that electron-donating substituents are performing better in terms of overall yield, although the regioselectivity does not change much. Is it possible to comment on the trend of the ligand substitutions? Authors should perform some kinetic experiments to see whether these substituents influence the oxidative addition step of the allylic alcohol.

Response: Thanks for the suggestion. The ligands with electron-deficient 4-fluorophenyl (**L2**) and electron-donating 4-methoxyphenyl (**L3**) are synthesized and evaluated. The ligand **L2** gave lower yield, and the ligand **L3** gave higher yield compared to ligand **L1** (phenyl), indicating that the ligand with electron-donating substituents performed better in terms of overall yield. This trend is consistent with that of substrate, that is, electron-donating substituent is beneficial to increase the yield of the reaction. These observations have been added to the text and Fig. 2 in the revised manuscript.

2) Line 98: The authors reported low regioselectivity for an open chain ketone **11** with secondary and tertiary α -carbons. Can the authors compute the transition barriers taking **11** as the model to compare the transition states and explain the low regioselectivity accordingly.

Response: Thanks for the suggestion. We calculated the transition barriers of the reaction with open-chain ketone **11**. The DFT results indicated that the transition barrier difference between the alkylations at secondary and tertiary α -carbons is only 1.1 kcal/mol, which is consistent with the observed low regioselectivity. These results have been added in the revised manuscript and the SI (Table S6 in SI).

3) Figure 6a: Can the authors explain the results of deuterium labelling experiments? Why percentage of deuterium incorporation is higher at the less-substituted end as compared to the more-substituted end, although the ratio of the more-substituted enolate to the less-substituted enolate is 70.1:29.9 (line 178).

Response: The ratio of enolates is determined by their thermodynamic stability, while the percentage of deuterium incorporation depends on the rate of keto-enol tautomerization. Generally, more-substituted enolate is more stable, and less-substituted α -carbon has faster keto-enol tautomerization rate.

4) Figure 5b: Authors mentioned regioselective and chemoselective allylic alkylation of ketones. Is that due to higher pka of those C-H's, since keto-enol tautomerism might not be possible then with the catalytic OH⁻. Some substrate having C-H with lower pka should be tried to check whether this chemoselectivity is a pka dependent phenomenon or not.

Response: Thanks for the suggestion. We conducted allylic alkylation of 1-phenylpentane-1,4-dione having two different secondary α -C-H bonds, and found that the reaction mainly occurred at the α -C-H bond with a lower pK_a . This result has been added in the revised SI (4d in SI) and a discussion was added in the revised manuscript.

5) Line 167: The figure references should be placed where it has been discussed

Response: Thanks for the suggestion. We have re-placed the figure references in the revised manuscript.

6) The authors should explain the significance of the data shown in table S2.

Response: Methyl ethyl ketone has four possible forms when it undergoes nucleophilic attack on the π -allylnickel intermediate in isopropanol solvent, including enol, enol anion and their isopropanol hydrogen-bonded forms. To determine which form of methyl ethyl ketone to attack π -allylnickel intermediate, we compared the free energy of each of these four forms. The calculation results indicated that the enol anion of methyl ethyl ketone is the main form to attack on the allyl nickel. This explanation has been added in Table S2 of SI.

7) Table S3: Why does TS1'' poses that high energy as compared to TS1? Why does the involvement of two molecules of isopropanol increase the transition barrier? Is it due to entropy-factor or space-constraint of the catalytic pocket?

Response: We think the reason for this is the space-constraint of the catalytic pocket. The narrow cavity of the catalytic pocket can only accommodate one isopropanol molecule to stabilize the **TS1** transition state. The bulky ligand results in a narrow cavity of the catalytic pocket.

8) Table S6: What is the difference between the optimized geometries Eb and Eb'? Is that a local minimum or that has been done through constraint optimization? That should be checked by frequency calculations and mentioned clearly.

Response: The difference between the optimized geometries Eb and Eb' comes from the relative positions (cis or trans to oxygen) of the methyl group of the alkyl chain. The geometric structures of Eb and Eb' presented in this paper are all local minimum structures. All local minimum structures reported in this paper were analyzed through vibration frequency analysis. These have been mentioned in Table S7 of SI.

Reviewer #3 (Remarks to the Author):

Zhou and colleagues reported an intriguing reactivity of the Ni/P-P catalytic system for γ -allylation of ketone at the sterically hindered position over the less sterically hindered favourable site. Because the reaction does not require any base, the only byproduct is water. A good scope of allylic alcohols and excellent regioselectivity may lead to this method being useful to a broader community. The linear selectivity obtained from branched alcohol is intriguing, indicating that the reaction proceeds via Ni- π -allyl species. The manuscript can be considered for publication; however, before acceptance, the following questions must be addressed.

1) Do other types of alcohols (not just allylic) react in the same way? Consider 1-octanol as an example. In doing so, we will rule out the alternative allylation pathway involving hydrogen borrowing chemistry.

Response: Thanks for the suggestion. We attempted to conduct a reaction by using 1-octanol under standard conditions, and did not obtain any product. This attempt has been added in the revised SI.

2) It would be fascinating if the author included terpenols (like geraniol, nerol, geranylgeraniol, etc.) in the substrate scope. This will be the first step in applying this method in the targeted synthesis.

Response: Thanks for the suggestion. We conducted the reaction using geraniol, and are unable to obtain the desired product. However, the reaction using geraniol derivatives afforded the desired product in high yield with excellent regioselectivity. This result has been added in the revised manuscript (Fig. 4, 6j).

3) How is stereoselectivity transformation accomplished? The question is why it is *E*-configuration rather than a combination of *E* and *Z*. If a mixture of *E* and *Z* allylic alcohols (1:1) is subjected to the reaction conditions, what would happen?

Response: Thanks for the suggestion. We conducted a reaction using an *E/Z* mixture (1:1) of 3-phenylprop-2-en-1-ol, and only obtained the product with *E*-configuration. This result has been added to the revised manuscript (Fig.4, 6b). Our hypothesis is that, in the reaction, *Z*- π -allylic nickel species transforms into the more stable *E*- π -allylic nickel species.

REVIEWERS' COMMENTS

Reviewer #1

< In private comments to the Editorial office, the reviewer said that the manuscript is ready for publication. >

Reviewer #3 (Remarks to the Author):

The authors address all the queries of the reviewer, and this fine manuscript is now acceptable for a publication. However, prior to the acceptance, the following information need to add in the text although the author mentioned that the details provided in the SI.

- As mentioned by authors, the reaction does not work with prenol or terpenols (although modified homogeranyl alcohol works!). It is understandable as the methyl group placed at the M-pi-allyl site may hinder the nucleophilic addition. It is important to provide that information in the manuscript as well.

A Point-By-Point Response to the Reviewers' Comments (NCOMMS-22-44784A)

Reviewer #1

Comment: In private comments to the Editorial office, the reviewer said that the manuscript is ready for publication.

Response: Thanks for the positive comment.

Reviewer #3 (Remarks to the Author):

The authors address all the queries of the reviewer, and this fine manuscript is now acceptable for a publication. However, prior to the acceptance, the following information need to add in the text although the author mentioned that the details provided in the SI.

As mentioned by authors, the reaction does not work with prenol or terpenols (although modified homogeryl alcohol works!). It is understandable as the methyl group placed at the M-pi-allyl site may hinder the nucleophilic addition. It is important to provide that information in the manuscript as well.

Response: Thanks for the suggestion. The information has been added to the text of revised manuscript.